# Impact of Futuristic Climate Variables on Weed Biology and Herbicidal Efficacy: A Review

Vipin Kumar [1], Annu Kumari [2], Andrew J. Price [3,*], Ram Swaroop Bana [4], Vijay Singh [5] and Shanti Devi Bamboriya [6]

1   Department of Plant Pathology, Physiology, and Weed Science, Virginia Polytechnic Institute and State University, Blacksburg, VA 24061, USA
2   Crop, Soil & Environmental Sciences Department, Auburn University, Auburn, AL 36849, USA
3   National Soil Dynamics Laboratory, Agricultural Research Service, United States Department of Agriculture, Auburn, AL 36832, USA
4   Senior Scientist Division of Agronomy, ICAR-Indian Agricultural Research Institute, New Delhi 110012, India
5   Eastern Shore Agricultural Research and Extension Centre, Virginia Tech, Painter, VA 23420, USA
6   ICAR-Indian Institute of Maize Research, Ludhiana 141004, India
*   Correspondence: andrew.price@usda.gov

**Abstract:** Our changing climate will likely have serious implications on agriculture production through its effects on food and feed crop yield and quality, forage and livestock production, and pest dynamics, including troublesome weed control. With regards to weeds, climatic variables control many plant physiology functions that impact flowering, fruiting, and seed dormancy; therefore, an altered climate can result in a weed species composition shift within agro-ecosystems. Weed species will likely adapt to a changing climate due to their high phenotypic plasticity and vast genetic diversity. Higher temperatures and $CO_2$ concentrations, and altered moisture conditions, not only affect the growth of weeds, but also impact the effectiveness of herbicides in controlling weeds. Therefore, weed biology, growth characteristics, and their management are predicted to be affected greatly by changing climatic conditions. This manuscript attempted to compile the available information on general principles of weed response to changing climatic conditions, including elevated $CO_2$ and temperature under diverse rainfall patterns and drought. Likewise, we have also attempted to highlight the effect of soil moisture dynamics on the efficacy of various herbicides under diverse agro-ecosystems.

**Keywords:** climate change; $C_3$ and $C_4$ weeds; moisture; $CO_2$ and temperature

## 1. Introduction

Climate change is the variation in global or regional climatic patterns that are measurable and persist for a prolonged time period. Since the beginning of the industrial revolution, the global climate has witnessed such deviations in climatic systems, mainly due to carbon dioxide emissions through the use of fossil fuels (https://climate.nasa.gov/causes/ accessed on 20 October 2021. The emission of greenhouse gases (GHGs) is increasing rapidly, and it is predicted that atmospheric $CO_2$ concentrations will reach 1000 PPM in the late 21st century with a simultaneous rise of about 2–4 °C in the Earth's average surface temperature [1]. The increase in global temperature, erratic distribution of rainfall, changed wind patterns, and extreme weather phenomena, such as severe storms, floods, and droughts, will likely occur due to past and continued emissions of greenhouse gases [2–5].

The changing climate is known to have serious implications on agriculture through its effects on food and feed crop quality and quantity; altered cropping systems and livestock production; and crop management practices for sustaining crop production under predicted climate change scenarios [6]. In addition to these impacts, a changing climate

also aggravates weed control [7]. Among the different biotic factors that reduce crop yield, weeds are the biggest threat and can result in yield losses of 34–37%, as compared to insects (18–29%), diseases (16–22%) and others (12%) [8–10]. In addition, non-controlled weeds also cause various direct and indirect losses to crops. Owing to their inherent phenotypic plasticity and vast genetic diversity, weeds can adapt to changing climatic conditions [11]. Climatic variables control plant flowering, fruiting, and seed dormancy; therefore, an altered climate can bring a shift in floral species composition of various ecosystems [12]. Weed species will adapt to the changing climate through migration, altered phenology and physiology, and growth behaviour. Higher temperatures and altered soil moisture conditions not only affect weed growth but also affects the effectiveness of herbicides in controlling weeds. Decreasing herbicide efficacy is problematic, as farmers across the globe rely heavily upon herbicides for the weed management of their crops, rangelands, and fallow lands [13]. Along with plant factors and herbicide chemical properties, herbicide efficacy is also dependent on climatic factors [14]. The absorption of herbicides in weeds is dependent on its interaction with plants leaves, soil, atmosphere, or the interface of plant leaf-soil-atmosphere [14]. Therefore, alterations in climatic conditions due to increasing $CO_2$ levels and their effect on the Earth's average surface temperature and shifting rainfall patterns can significantly affect the growth of weed plants and the efficacy of herbicides.

The use of herbicides is widely adopted and is an efficient weed management practice; therefore, herbicide use in the future requires an understanding of the consequences of the changing climate on the growth of weed plants and the efficacy of herbicides. The main objective of this manuscript was to analyze the probable effects of altered climate on the growth and biology of weed species and the efficacy of herbicides.

## 2. Impact of Climate Change on Weed Biology

### 2.1. Photosynthetic Cycles

The differential reactions of $C_3$ and $C_4$ plants to changed climatic conditions necessitate a deeper understanding of $C_3$ and $C_4$ photosynthetic cycles in weeds (Table 1).

**Table 1.** Major differences between $C_3$ and $C_4$ plants.

| Characteristics | $C_3$ Plants | $C_4$ Plants |
|---|---|---|
| $CO_2$ acceptor | Ribulose-1,5-bisphosaphate | Phosphoenolpyruvate |
| Enzyme | Rubisco | PEP carboxylase |
| Extent of photorespiration | High | Negligible |
| Saturation light intensity | 1000–4000 foot candle | Hardly reaches saturation even in full sunlight |
| $CO_2$ compensation point | 50–150 ppm | 0–10 ppm |
| Operating $CO_2$ fixation pathway | Only $C_3$ pathway | Both $C_3$ and $C_4$ pathway |
| Optimum temperature for growth | 10–25 °C | 30–45 °C |
| Preferred climatic condition | Dominant in temperate regions | Dominant in tropical and sub-tropical areas |
| Abundance | Found in both angiosperms and gymnosperms | Found only in angiosperms |

Source: Das [8].

### 2.1.1. $C_3$ Plants

In $C_3$ plants, the enzyme which acts as an acceptor of $CO_2$ is Ribulose 1,5-bisphosphate carboxylase oxygenase (Rubisco). Rubisco can perform both carboxylation and oxidation functions. The extent of carboxylation or oxidation depends upon the ratio of atmospheric concentration of $CO_2$ to $O_2$ (Table 1). If $CO_2$ level increases, then carboxylation is favored, but if $CO_2$ concentration decreases then oxidation is favored. For plants, oxidation is an energy-wasteful process, whereas carboxylation leads to the production of photosynthates. Increased atmospheric $CO_2$ concentration results in a corresponding increase in net photosynthesis, hence it is beneficial to $C_3$ plants. However, a temperature increase is detrimental to $C_3$ plants because the photorespiration rate increases when the temperature exceeds 25 °C [15]. In addition, drought and water stress conditions disfavor $C_3$ weeds because $C_3$ plants have low water use efficiency compared to $C_4$ plants. Therefore, $C_4$ plants have an advantage over the plants with $C_3$ photosynthetic pathways [12].

### 2.1.2. $C_4$ Plants

In $C_4$ plants, Phospho-enol-pyruvate carboxylase (PEP) is the $CO_2$ acceptor enzyme. It can catalyse only one reaction, carboxylation. In $C_4$ plants, the $CO_2$ supply mechanism to PEP is controlled internally, hence $C_4$ plants do not depend much on atmospheric $CO_2$ for photosynthesis. Also, $C_4$ plants become saturated in terms of $CO_2$ fixation at a concentration of around 360 ppm, thus becoming less sensitive to increased $CO_2$ [16]. However, an increased temperature enhances the rate of photosynthesis in $C_4$ plants, with a negligible loss of energy through photorespiration [17]. In addition, the water-use efficiency of $C_4$ plants is higher compared to $C_3$ plants, thus weeds with a $C_4$ mechanism are likely to be more competitive compared to $C_3$ plants under water-stressed ecologies and increased temperature conditions [12]. A list of different $C_3$ and $C_4$ plants is presented in Table 2. Some of these $C_4$ weeds are more competitive than others, as higher phenotypic plasticity and huge genetic diversity influence sensitivity towards the changing climatic conditions on the weeds [18].

**Table 2.** Major $C_3$ and $C_4$ weeds compete with crops.

| $C_3$ Weed Species | $C_4$ Weed Species |
| --- | --- |
| *Avena fatua* | *Amaranthus viridis* |
| *Abuliton theophrasti* | *Boerhavia diffusa* |
| *Ageratum conyzoides* | *Cynodon dactylon* |
| *Bidens pilosa* | *Cyperus rotundus* |
| *Chenopodium album* | *Dactyloctenium aegypticum* |
| *Cirsium arvense* | *Digitaria sanguinalis* |
| *Commelina benghalensis* | *Echinochloa crus-galli* |
| *Convolvulus arvensis* | *Elusine indica* |
| *Eichhornia crassipes* | *Fimbristylis miliacea* |
| *Phalaris minor* | *Ischaemun rugosum* |
| *Striga asiatica* | *Portulaca oleracea* |
| *Rumex dentatus* | *Sorghum halpense* |
| *Xanthium strumarium* | *Trianthema portulacastrum* |

Source: Das [8].

### 2.2. Basic Principles of Weeds' Response to an Increased $CO_2$ and Temperature Climate

The continuous survival and existence of a weed species requires adaptability to change in climatic conditions [19]. Weeds can adopt three different adaptability mechanisms for survival [20,21].

Migration: This mechanism is also called range shift, and it includes the movement/shift of weed species from one place to another, more favourable place. Several anthropogenic activities, such as using crop seeds and organic manures having weed-seed contamination, movement of farm equipment, transport of crop produce having weed seed mixed with it, also contribute to the dispersal of weed seeds [8,22,23].

Acclimation: This mechanism may be called a niche shift, and it refers to the reactions of plant species to the altering conditions through modifications in their phenotype, but these phenotypic modifications are not evolutionary and are non-heritable [24,25].

Adaptation: This mechanism is known as trait shift, and it includes various heritable changes which occur in a plant species in response to their altered environments. This is a type of natural selection, with the development of new characteristics and an optimizing the existing ones [26–28].

### 2.3. Weed Response to Elevated Atmospheric $CO_2$ Concentration

Increasing $CO_2$ concentrations significantly affect weed growth, development, and biology, as well as crop–weed interference. Various studies carried out to compare the effect of escalated $CO_2$ concentrations on $C_3$ and $C_4$ weeds show that increased $CO_2$ levels favor the growth and development of $C_3$ species over the weeds which have $C_4$ pathways [29,30].

The influence of increasing $CO_2$ levels on the biomass of important $C_3$ and $C_4$ weeds is presented in Table 3.

**Table 3.** Effect of elevated $CO_2$ concentrations on various $C_3$ and $C_4$ weed species.

| $C_3$ Species | Biomass (% of Ambient) | $C_4$ Species | Biomass (% of Ambient) |
|---|---|---|---|
| *Ambrosia artemisiifolia* | 110–133 | *Amaranthus retroflexus* | 96–141 |
| *Datura stramonium* | 174–272 | *Cyperus rotundus* | 102 |
| *Elymus repens* | 164 | *Digitaria ciliaris* | 106–121 |
| *Rumex acetosella* | 131 | *Echinochloa crusgalli* | 95–159 |
| *Cirsium arvense* | 121 | *Elusine indica* | 102–121 |
| *Chenopodium album* | 100–155 | *Sorghum halepense* | 56–110 |
| *Crotolaria spectabilis* | 167 | *Paspalum pilcatulum* | 108 |
| *Parthenium hysterophorous* | 953 | *Cenchrus ciliaris* | 104 |
| *Sesbania cannabina* | 100 | *Setaria faberi* | 9 |
| | | *Setaria virdis* | 19 |
| | | *Sorghum halapense* | 8 |
| | | *Panicum dichotomiflorum* | 24 |

Source: Modified from Patterson [29]; Navie et al. [30].

The roots and shoots of different plant species also respond differently to elevated $CO_2$ [31]. For example, both roots and shoots of *Cirsium arvense* (L.) Scop. accumulated more biomass when grown under increased $CO_2$ conditions, but the accumulation in root biomass was higher compared to shoot biomass [32]. Increasing $CO_2$ levels also act as a selection factor for the evolution of plants. In an experiment, Ziska [33] grew two populations of wild oat i.e., old and new populations. Old population seeds were collected in 1966 and 1967 and the new population seeds were collected in 2014. These seeds were grown at 315 ppm and 420 ppm $CO_2$ to represent the $CO_2$ concentrations of the mid-20th century and current $CO_2$ levels. In this study, the new population showed a significant increase in leaf, stem, and total plant weight under increased $CO_2$, whereas there was no significant increase for the old population. Climate-induced selection favors weed populations more than crops. In the same study, cultivated oats were also grown along with the new wild oat population and it was found that under ambient $CO_2$ conditions, cultivated oats had a higher leaf area and above-ground biomass, whereas, under increased $CO_2$, both leaf area and above-ground biomass were higher for the new wild oat population. Relative growth rate, net photosynthesis rate, and biomass accumulation of two $C_3$ weeds, *Euphorbia heterophylla* L. and *Commelina diffusa* Burm. f. increased by 80.5%, 16.7%, 143% and 60.5%, 9.5%, 108%, respectively, under increased $CO_2$ conditions as compared to normal $CO_2$ [34]. Under increased $CO_2$ concentrations, similar increases in pod and seed yields, and biomass accumulation of the $C_3$ weed *Abutilon theophrasti* Medik. were reported [35]. Elevated $CO_2$ has resulted in increases in leaf area by 40%, 67%, and 24%, and biomass by 39%, 83%, and 59% in *Chloris gayana* Kunth, *Eragrostis curvula* (Schrad.) Nees, and *Paspalum dilatatum* Poir, respectively, while there was a decrease in leaf area by 34% and biomass by 5% in *Sporobolus indicus* (L.) R.Br. [36]. An increase in *Chenopodium album* L. biomass was observed even up to a 1500 ppm $CO_2$ concentration. However, a further increase in $CO_2$ concentration to 3000 ppm reduces biomass accumulation in *C. album* but still, the biomass accumulation remained higher than that at 350 ppm [37]. $C_4$ weeds can show an increase in their growth under increased $CO_2$ if they are subjected to water stress [38]. The iomass and height of water-stressed *Amaranthus retroflexus* L. increased by raising the $CO_2$ concentration [38]. Therefore, the response of weed plants is not solely dependent upon $CO_2$ concentrations, but also depends upon various other climatic factors.

### 2.4. Weed Response to Increased Temperature

Temperature is a crucial factor in determining the presence or absence of a particular weed species in a particular region [39]. Any alteration in temperature is likely to change the growth, development, and distribution pattern of weed plants. In general, increased

temperature favors $C_4$ weeds over $C_3$ weeds, mainly due to high photorespiration in $C_3$ plants under higher temperatures [14]. Higher temperatures accelerate canopy growth and root proliferation in $C_4$ plants [40]. Infestations of various $C_4$ weeds, such as *Panicum dichotomiflorum* Michx., *Digitaria* sp., *Amaranthus retroflexus*, *Setaria* sp., and *Sorghum halepense* (L.) Pers. are likely to increase further northward in the northern hemisphere and southward in the southern hemisphere [41,42]. The survival of winter annual weeds will likely increase under mild and wet winter conditions, while warm summers and long growing seasons will favor the growth of thermophilic summer annuals, allowing them to grow well in relatively colder regions [43,44]. *Datura stramonium* L., a weed that exhibits high growth under high temperatures [45], is expected to become a more aggressive competitor in the face of altered climatic conditions [46].

Lee [47] reported that emergence and flowering of *Chenopodium album* advanced by 26 and 50 days, respectively by an increase of 4 °C in temperature, while this advancement in the emergence and flowering time was 35 and 31.5 days, respectively for *Setaria viridis* (L.) P. Beauv. Bir et al. [48] reported an increase in plant height as well as the leaf area of different weed species in paddy fields under elevated temperatures (Figure 1).

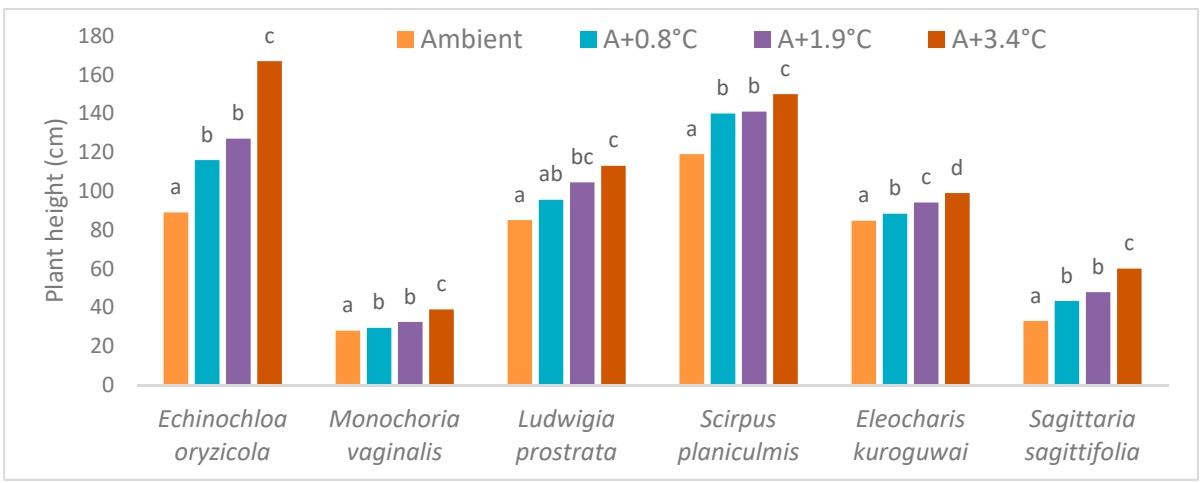

**Figure 1.** Plant height (cm) of different weeds in a paddy field under increased temperatures (Source: Modified from Bir et al. [48]). Different lower cases indicate significant differences among treatments.

### 2.5. Weed Response to Changed Rainfall Pattern and Drought Spells

Changes in soil moisture availability and rainfall patterns, along with elevated aridity and warmer climates, affects the growth, development, survival, and distribution of not only crop plants but also weeds. Drought conditions may harm $C_4$ weeds to a lesser degree compared to $C_3$ weeds [38], which can be attributed to the high water-use efficiency of $C_4$ weeds. However, in excess moisture conditions, $C_3$ weeds, such as *Rhamphicarpa fistulosa* (Hochst.) Benth. will be favored, while drought-like situations will favour $C_4$ and parasitic weeds such as *Striga hermonthica* (Del.) Benth. [49].

Drought conditions enhance competition by *Amaranthus. theophrasti* and *Anoda cristata* (L.) Schtdl. with cotton [50], while Mortensen and Coble [51] found that yield loss in soybean because of *Xanthium strumarium* L. infestation was higher under optimum soil-moisture conditions (~29%) than moisture-stressed conditions (~12%). Similarly, *Cirsium. arvense* is more competitive to wheat under high rainfall conditions [52]. Some crop plants cannot survive water-stress conditions, but weeds can survive under such conditions. Rice plants were unable to survive under low soil-moisture conditions (25% and 12.5% of field capacity), whereas *Amaranthus spinosus* L. and *Leptochloa chinensis* (L.) Nees were not only able to survive, but also produced a sizeable number of branches/tillers at similar soil-moisture levels [53].

### 2.6. Weed Response to the Interaction of Climatic Variables

Climate is complex, and does not consist of only one or two variables, but rather consists of numerous variables functioning together. As stated earlier, increasing $CO_2$ levels favour the growth and development of $C_3$ species by increasing the net photosynthesis [12], but this $CO_2$ fertilization can be altered by some other climatic variables. For instance, increased temperatures and drought-like conditions favor $C_4$ plants in comparison to $C_3$ plants [12]. It has been predicted that both $CO_2$ concentrations and temperatures are expected to further increase in future years [54]. Despite this prediction, very few experiments have been conducted to study the impact of the simultaneous increase in $CO_2$ and temperature on weed species. Most of the experiments have been conducted either under increased $CO_2$ levels or increased temperatures, but the results of these studies may be different if both these parameters are taken into consideration. Valerio et al. [55] found that increasing $CO_2$ from 400 to 800 ppm reduced the crop losses due to *Chenopodium album* and *Amaranthus retroflexus* in tomato from 33% to 32%, but when both CO2 concentration and temperature were increased from 400 to 800 ppm and 21/12 °C to 26/18 °C day/night respectively, then crop losses due to weed infestation increased from 55% to 61%. Lee [47] reported that biomass accumulation and seed production of *Chenopodioum album* ($C_3$ weed) decreased when the atmospheric temperature was increased by 4 °C, but a simultaneous increase in temperature by 4 °C and $CO_2$ by 1.8 times of the ambient levels increased biomass accumulation and seed production. However, similar treatments applied to *Setaria viridis* ($C_4$ weed) did not show any individual effect of increased temperature on biomass accumulation and seed production, but a significant interaction effect of both temperature and CO2 elevation was observed (Figure 2). Another experiment conducted in the Philippines evaluated the effect of CO2 concentration and temperature on the competitiveness of *Echinochloa glabrescens* Munro ex Hook. f. with rice crop. This research indicated that an elevation in CO2 concentration to 594 ppm provided a competitive edge to rice but when temperature was increased from 27/21 °C (day/night) to 37/29 °C (day/night), *Echinochloa glabrescens* ($C_4$ weed) was more competitive to rice [56].

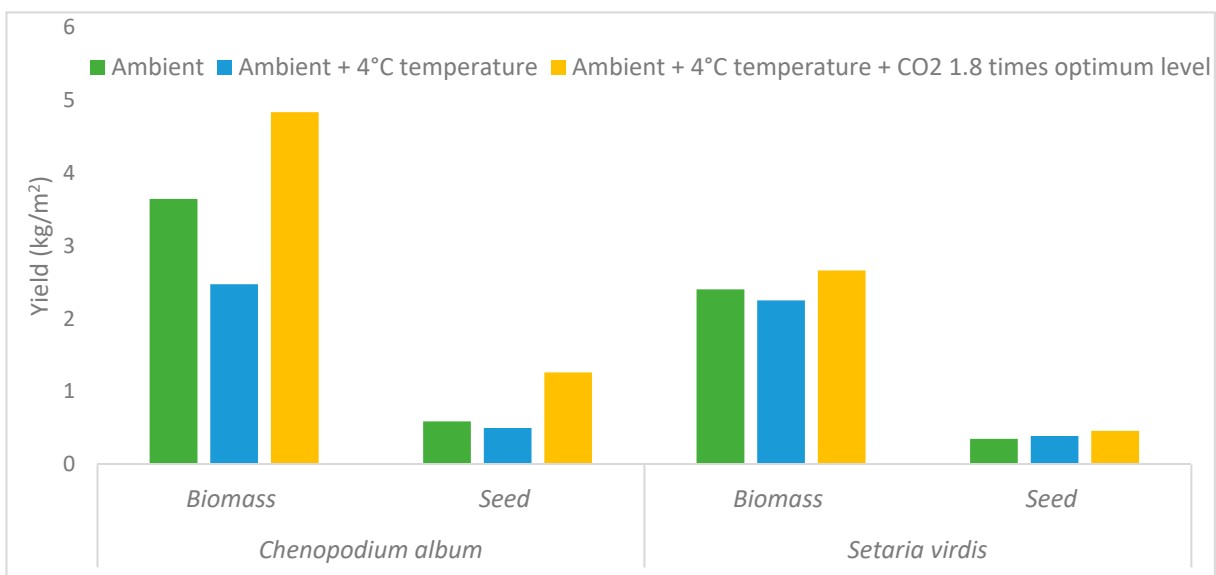

**Figure 2.** Effect of climatic factors on biomass (kg/m$^2$) and seed yield (kg/m$^2$) of *Chenopodium album* ($C_3$) and *Setaria virdis* ($C_4$) (Source: Modified from Lee [47]).

Relatively few studies have been conducted on the interaction of moisture and $CO_2$. Normally increased $CO_2$ concentration favours $C_3$ weeds but under water-stressed conditions elevated $CO_2$ levels may also favor the growth of $C_4$ weeds [38]. Improvements in the water-use efficiency of different $C_3$ and $C_4$ weeds by doubling the $CO_2$ concentration from

300 to 600 ppm were reported by Carlson and Bazzaz [57]. By doubling $CO_2$ concentration from 300 to 600 ppm, the water-use efficiency of crops such as sunflower, soybean, and maize increased by 55%, 48%, and 54%, respectively, while the water-use efficiency of weed species such as *Ambrosia artemisiifolia* L., *A. theophrasti*, *D. stramonium* and *A. retroflexus* increased by 128%, 87%, 84%, and 76%, respectively.

Furthermore, growth of *Digitaria ciliaris* (Retz.) Koeler, *Echinochloa crus-galli* (L.) P.Beauv., and *Eleusine indica* (L.) Gaertn. ($C_4$ weeds), as well as soybean ($C_3$ crop) is favored by the rising $CO_2$ concentrations because of improvements in their water-use efficiency under drought [58]. The IPCC [59] predicts changes in temperature and available moisture due to changing climates, which will affect the germination and temporal and spatial emergence of weed seeds and seedlings [60]. Weed-seed dormancy, one of the crucial factors governing the emergence of weed species, is believed to be broken faster due to the higher availability of moisture and high temperature conditions [61,62]. Likewise, nutrient supply also influences crop–weed interference. Zhu et al. [63] found that under optimum nitrogen level increasing the $CO_2$ concentration by 200 ppm proportionately increased rice biomass as compared with *E. crus-galli*, while under elevated $CO_2$ level and sub-optimum nitrogen supply, the competitive ability of rice decreased.

## 3. Changing Climate and Efficacy of Herbicides

Among various practices to manage weeds e.g., mechanical, cultural, chemical, and biological methods—chemical weed control is a widely adopted method owing to its economic advantages and ease of application [13]. However, in the past few decades, the concept of integrated weed management has become a popular method to reduce the dependency on herbicides due to the development of herbicide-resistant weeds [64] and various other challenges created by herbicide use. Given the role of herbicides in the management of weeds and reducing crop-yield losses, it is very important to evaluate how the efficacy of herbicides will be affected by changing climatic conditions, including $CO_2$ concentration, temperature, wind, relative humidity, moisture availability, and light [14].

### 3.1. Increased $CO_2$ Levels and Herbicide Efficacy

The efficacy of herbicides may change due to rising $CO_2$ levels, mainly because of changes in physiological processes. Increased $CO_2$ levels can decrease stomatal conductance by up to 50% in some plant species [65]. The activity of foliage-applied herbicides might be altered by decreased stomatal conductance [66]. Furthermore, increased leaf thickness and the reduced count of open stomata due to high levels of $CO_2$ may reduce the efficacy of the post-emergence herbicides directly absorbed by the plants. For example, foliar uptake of glyphosate is reduced in high $CO_2$ environments because of thickened leaves, reduced stomatal conductance, and the number of stomatal openings, thereby reducing its efficacy [67–69]. In addition, the decreased requirement of aromatic amino acids (tryptophan, phenylalanine, and tyrosine) owing to the low-protein content in plants under increased $CO_2$ conditions [70,71] further reduces the effectiveness of glyphosate because glyphosate inhibits the activity of 5-EPSPS enzyme, which catalyzes the synthesis reaction of these aromatic amino acids [72].

Various studies have highlighted that the effects of altered $CO_2$ on the amino acid inhibitor mode of action herbicides is weed species-specific. The application of glyphosate on weeds grown under elevated $CO_2$ conditions resulted in a higher survival rate of *C. gayana*, *E. curvula*, and *P. dilatatum* as compared to the plants grown under normal $CO_2$ conditions, while there was no effect of increased $CO_2$ on the survival of *Sporobolus indicus* (L.) R. Br. [36]. Ziska et al. [72] reported that the application of glyphosate under increased $CO_2$ concentrations did not affect the growth of *C. album*, whereas growth of *A. retroflexus* was reduced under both normal and elevated $CO_2$ conditions. The efficacy of imazamethabenz in controlling *Avena fatua* L. increased by 15.7% by doubling the normal $CO_2$ concentration, while metsulfuron-methyl efficacy on *A. retroflexus* declined by 4.6%, and there was no change in imazethapyr efficacy on *Stellaria media* (L.) Vill. [73]. Increased

$CO_2$ levels can provide higher carbon resources, resulting in more raw materials for fatty acid synthesis, which reduces the efficacy of fatty-acid-synthesis-inhibitor herbicides [74]. Higher $CO_2$ levels stimulate the growth of weeds, reducing the time a weed plant spends in the seedling stage during which weeds are highly susceptible to herbicidal action, thereby reducing the time window for herbicide application [7]. Rising $CO_2$ concentrations stimulate the growth of roots more than the shoots, and this has serious implications in the control of weed plants which propagate through underground vegetative parts, such as *C. arvense*, in which the increased growth of underground parts has resulted in the decreased efficiency of glyphosate [32]. Herbicide-resistant and susceptible weeds respond differently to herbicides under increased $CO_2$ concentrations, as indicated by the study conducted by Refatti et al. [74], in which it was found that under normal $CO_2$ conditions, both multiple resistant and susceptible genotypes of *Echinochloa colona* (L.) Link were killed equally after the application of cyhalofop-butyl, whereas under elevated $CO_2$ conditions, cyhalofop-butyl was able to kill almost all plants of the susceptible genotype, while the control of the multiple-resistance genotype decreased from nearly 100% to less than 80%.

### *3.2. Temperature and Herbicide Efficacy*

The efficacy of herbicides can be influenced directly and indirectly by temperature. Temperature controls numerous physiological processes in plants, such as respiration, photosynthesis, protoplasmic streaming, and phloem translocation [14]. Furthermore, temperature also regulates the growth and development of plants. Therefore, the translocation and penetration of herbicides inside the plants are indirectly affected by alterations in these processes. The diffusion of herbicides, physicochemical characteristics of spray solutions, and viscosity of cuticular waxes can be directly affected by temperature, consequently affecting herbicide efficacy [75]. Elevated temperatures can enhance the absorption and translocation of herbicides, which can increase the efficacy of herbicides. However, under higher temperatures, the metabolism of herbicides is also rapid, which can reduce the performance of herbicides on target weed species [76]. The increased absorption and translocation of herbicides under higher temperatures can cause phytotoxic effects on crop plants. For example, higher [14]C-glyphosate translocation to meristematic tissues in glyphosate-resistant soybean at 35 °C as compared to 15 °C, causes glyphosate injury at elevated temperatures [77]. Godar et al. [78] reported a reduction in the effectiveness of mesotrione on *Amaranthus palmeri* S. Watson at higher temperature (40/30 °C day/night), compared to the optimum temperature of 32.5/22.5 °C day/night and lower temperatures of 25/15 °C day/night. Additionally, Godar et al. [78] found an increase in median effective dose ($ED_{50}$) value (g a.i. $ha^{-1}$) of mesotrione for plant height reduction, visual injury, chlorophyll reduction, photochemical efficiency of PSII ($F_V/F_M$) and mortality (Figure 3).

Increasing temperatures enhance the metabolism of herbicides, which decreases the efficacy of herbicides and increases the risk of the development of non-target site herbicide resistance in weed species [79]. The survival of four grass weeds (*Avena sterilis* L., *Alopecurus myosuroides* Huds., *Lolium rigidum* Gaudin and *S. viridis*) increased from 0% to 80% by an increase in temperature from 16/10 °C day/night to 34/28 °C day/night [79]. Under normal temperatures (23/35 °C, night/day), the effectiveness of cyhalofop-butyl on susceptible genotypes was numerically higher (injury nearly 90%) than multiple-resistant genotypes (injury nearly 80%) of *E. colona*, while under higher temperatures (26/38 °C, night/day), the injury percentage remained statistically similar to the susceptible genotype but the injury percentage in the multiple-resistant genotype decreased to nearly 50% [74]. This increase in resistance due to higher temperatures can be attributed to the upregulation of the genes involved in the abatement of abiotic stress and detoxification of herbicides [80,81]. In multiple-resistant genotypes, a hypersensitive response is also observed which results in rapid killing of the herbicide-exposed cells and quick recovery when combined with other integral coping mechanisms [80]. In contrast, a decrease in resistance against glyphosate due to its enhanced translocation was observed in *A. artemisiifolia* and *Ambrosia trifida* L. at higher temperatures [82]. Additionally, higher temperatures decreased $ED_{50}$ (herbicide

dose for 50% control) and $GR_{50}$ (herbicide dose for 50% growth reduction) values for 2,4-D and glyphosate (Table 4) against *A. artemisiifolia* and *A. trifida* [82]. Studies have reported that in general, herbicide droplets dry rapidly under higher temperatures, which results in the decreased uptake of herbicides [83]. The volatility of various herbicides, such as synthetic auxins, is affected by temperature, which causes injury to different non-target broadleaf plants through vapor drift [84]. Behrens and Lueschen [85] found that visual injury in soybean due to the drift of dicamba increased from 0% to 40% when the temperature increased from 15 °C to 30 °C.

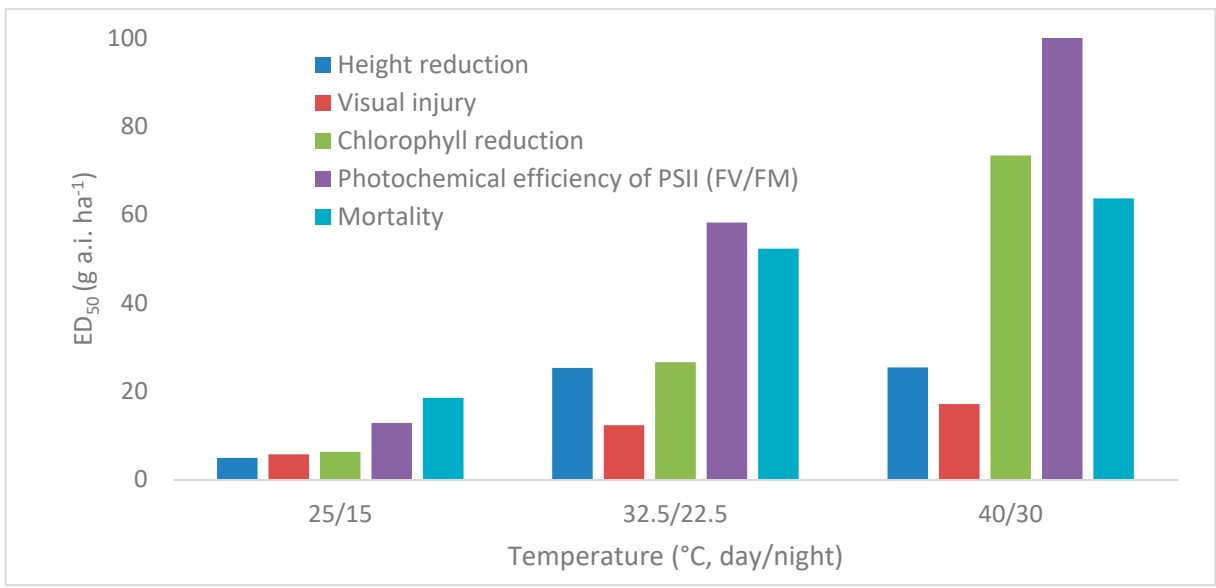

**Figure 3.** $ED_{50}$ value (g a.i. ha$^{-1}$) of mesotrione for different parameters under varying temperatures (Source: Modified from Godar et al. [78]).

**Table 4.** $ED_{50}$ (g ae ha$^{-1}$) and $GR_{50}$ (g ae ha$^{-1}$) dose of different herbicides for different weed species under increased temperatures.

| Weed Species | Herbicide | | Low Temperature (20/11 °C Day/Night) | | High Temperature (29/17 °C Day/Night) | |
|---|---|---|---|---|---|---|
| | | | $ED_{50}$ | $GR_{50}$ | $ED_{50}$ | $GR_{50}$ |
| *Ambrosia artemisiifolia* | 2,4-D | | 187 | 20 | 61 | 17 |
| | Glyphosate | Susceptible biotype | 437 | 45 | 130 | 39 |
| | | Resistant biotype | 2821 | 323 | 1307 | 306 |
| *A. trifida* | 2,4-D | | 71 | 15 | 13 | 25 |
| | Glyphosate | Susceptible biotype | 119 | 59 | 62 | 49 |
| | | Resistant biotype | 1429 | 349 | 1164 | 218 |
| | | | **Low Temperature (17.5/7.5 °C Day/Night)** | | **High Temperature (32.5/22.5 °C Day/Night)** | |
| | | | $ED_{50}$ | $GR_{50}$ | $ED_{50}$ | $GR_{50}$ |
| *Kochia scorpia* | Glyphosate | Pratt county population | 39 | 34 | 173 | 171 |
| | | Riley county population | 36 | 46 | 186 | 187 |
| | Dicamba | Pratt county population | 52 | 21 | 106 | 73 |
| | | Riley county population | 105 | 46 | 410 | 225 |
| | | | **Low Temperature (18 °C)** | | **High Temperature (32 °C)** | |
| | | | $GR_{50}$ | | $GR_{50}$ | |
| *Amaranthus tuberculatus* | Mesotrione | | 3.6 | | 22.6 | |
| *Digitaria sanguinalis* | Mesotrione | | 11.1 | | 82.1 | |

Source: Modified from Johnson and Young [76]; Ganie et al. [82].

Soil temperature is one of the important climatic factors which affect the degradation of herbicides. As the soil temperature increases, the rate of herbicide degradation also increases due to enhanced microbial and chemical breakdown [86,87]. The half-life of S-metolachlor decreased from 64.8 to 38.9, 26.3, and 23.7 days when the soil temperature increased from 10 to 15, 25, and 35 °C, respectively, while keeping the soil moisture constant [87]. Likewise, the half-life of isoxaflutole decreased from 13.9 to 3.3, 1.3, and 0.8 days when the soil temperature increased from 5 to 15, 25, and 35 °C, respectively [88].

Along with temperature, soil type also affects volatilization loss of triallate herbicide, as indicated in a study by Atienza et al. [89], which demonstrated that increasing the temperature from 5 °C to 25 °C increased the losses of triallate herbicide by 14% to 60% in sandy soils and by 7% to 41% in loamy soils. The percentage of initial atrazine concentrations present after 60 days of application in clay soil was 73% and 80% at 30/16 °C and 16/8 °C (day/night), respectively, whereas in loamy sand the concentration was 45% and 70% at 30/16 °C and 16/8 °C (day/night), respectively [90].

### 3.3. Precipitation, Soil Moisture, and Herbicide Efficacy

The changing climate will affect the temporal and spatial distribution of rainfall, and the frequency of extreme events, such as drought and floods, is likely to increase [2]. Rainfall affects the uptake of herbicide by washing the herbicide droplets from the leaf surface or by reducing the concentration of herbicides due to dilution [91]. Rainfall controls the soil moisture, which in turn affects herbicide efficacy. The adsorption of herbicides is more pronounced in dry soil [92], while heavy rainfall increases the leaching of herbicides [93]. Herbicide efficacy is adversely affected by water-stressed conditions because of reduced translocation and lower transpiration rates [94,95]. Zhou et al. [96] found that the efficiency of glyphosate in controlling *A. theophrasti* reduced under stress. Even the addition of some adjuvants was unable to improve the efficacy of glyphosate under stress conditions (Table 5).

**Table 5.** Fresh weight reduction (%) of *Abutilon theophrasti* due to glyphosate application with different adjuvants.

| Adjuvant | Source of Stress | | |
|---|---|---|---|
| | None | Drought | Flooding |
| None | 84 | 46 | 50 |
| Ammonium sulphate | 90 | 58 | 60 |
| Ammonium nitrate | 85 | 50 | 48 |
| Non-ionic surfactant | 89 | 60 | 62 |
| Methylated seed oil | 84 | 45 | 50 |
| Petroleum oil concentrate | 83 | 50 | 53 |

Source: Modified from Zhou et al. [96].

Similarly, Pereira [97] reported a decreased in efficacy of Acetyl CoA Carboxylase (AC-Case) inhibitors in water-stressed *Urochloa plantaginea* (Link) R.D.Webster. The efficacy of preemergence-applied pethoxamid also decreased under low soil-moisture conditions [98]. Under water-stressed conditions, the uptake of soil-applied herbicides through roots was reduced, because of the reduced solubility and movement of herbicide in the soil [99,100]. For the efficient translocation of systemic herbicides, weed plants must be actively growing, therefore water-stressed weed plants are difficult to control with post-emergence herbicides [101]. Soil moisture also influences leaf orientation, where plants leaves are tilted downwards under reduced soil moisture conditions, thereby reducing the uptake of foliar-applied herbicides [96]. Moreover, soil moisture also affects the degradation of soil-applied herbicides by regulating chemical and biological activities [86,87].

*3.4. Interaction of Climatic Variables and Herbicide Efficacy*

Extensive research has been carried out to evaluate the individual effect of various climatic variables on the efficacy of different herbicides, but only a few studies have been conducted that focused on the combined influence of various climatic variables. Elevated temperatures and $CO_2$ levels can modify leaf characteristics, such as increases in leaf thickness or alter the viscosity of cuticular waxes, which subsequently reduces the absorption of herbicides [102]. Increased temperatures and $CO_2$ levels can intensify the development of non-target site weed resistance [74]. Moreover, high temperatures and $CO_2$ concentrations affect photosynthesis activity, thereby impacting photosynthesis-interfering herbicides [14]. Under increased temperatures and $CO_2$ levels, the translocation of glyphosate into shoot apical meristems and younger leaves was increased, which resulted in the quick death of affected leaves and decreased translocation to other plant tissues [103]. Additionally, Matzrafi et al. [103] found the percentage survival of *C. album* and *Conyza canadensis* (L.) Cronq. following glyphosate application was 8%, 34%, and 61%, and 19%, 41%, and 64% under increased $CO_2$ levels and temperatures, respectively, as compared to normal temperatures and $CO_2$ levels (3% and 12%, respectively). In contrast, elevated atmospheric temperatures combined with increased relative humidity is favorable for efficient weed control by amino acid-inhibitor herbicides [104].

**4. Conclusions**

Climate change can affect weed dynamics, weed biology, and their management by influencing various physiological and biochemical processes. Weed plants are responding under the changing climate, which has serious consequences in their management, specifically through herbicides. The altered moisture content with higher temperature and $CO_2$ concentrations affects weed growth and impacts the effectiveness of herbicides in controlling weeds. Most studies reported a decrease in herbicide efficacy under changed climatic factors, while a few studies also reported an increase in herbicidal efficacy. In addition, most of the studies were done by altering a single climatic variable, but climate is a complex phenomenon; therefore, there is a need to study the impact of the complex interactions of various climatic factors and assess their long-term effects on agricultural systems. Additionally, it is essential to further investigate the effectiveness of various herbicides against different weed-species groups under variable climatic conditions. Moreover, future weed management research will need to consider the evolution of weed plants in addition to the complexities of environmental factors. Furthermore, weed species-specific management protocols for diverse soil, tillage, management, and farm typologies will need to be investigated in cropping system modes to develop pragmatic solutions for changing scenarios.

**Author Contributions:** Conceptualization, V.K. and A.K.; methodology, V.K.; software, A.K.; validation, A.J.P., V.S. and R.S.B.; formal analysis, A.K.; investigation, A.J.P.; resources, S.D.B.; data curation, V.S.; writing—original draft preparation, V.K.; writing—review and editing, A.K.; visualization, R.S.B.; supervision, V.S.; project administration, A.J.P.; funding acquisition, A.J.P. All authors have read and agreed to the published version of the manuscript.

**Funding:** This research received no external funding.

**Data Availability Statement:** The data that support the findings of this paper are available from the corresponding author upon request.

**Conflicts of Interest:** The authors declare no conflict of interest.

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
