# Peer review of "Impact of Futuristic Climate Variables on Weed Biology and Herbicidal Efficacy: A Review"

_agronomy, doi:10.3390/agronomy13020559_

Round 1

Reviewer 1 Report (Previous Reviewer 3)

The Authors managed to improve the Manuscript according to the Reviewers' comments. 

I suggest the Editor to accept the Manuscript in the presented form form possible publication in the Agronomy journal.

Author Response

no specific comments from reviewers 

Reviewer 2 Report (Previous Reviewer 1)

In my opinion, the re-reviewed manuscript has been sufficiently revised and can be published in Agronomy.

January 20, 2023.

Author Response

no specific comment from reviewers, manuscript met criteria according to their opinion 

Reviewer 3 Report (New Reviewer)

Comments: Impact of futuristic climate variables on weed biology and herbicidal efficacy: a review

The paper is interesting but there are some issues that need authors' attentions. I give my assessment of the manuscript section wise:

Abstract: the background is presented well and objectives are clear, however, the authors did not mention about the methods or approach, even  though it is a review, but must narrate the coverage of review in terms of scope, time period and nature of litrature reviewed along with source. In addition, 1-2 sentences on key findings could also make it attractive/insightful for the journal readers. Keywords also need a revisit as per my understanding.

Introduction: 

It is well-presented and contextualized. The background of the topic along with justification and nature of scope of the work is effectively taken up in this section. Section 2.1 needs a revisit as it has been overlooked to write its name. Both the sections 2 and 3 are a sort of results and discussion but as pointed out earlier, it would have been better if authors could clarify the selection of studies highlighted/referred in these sections. One would be more interested in the selection of literature more systematically rather than randomly. However, the selected literature is relevant and suited to the topic of the paper. One of the main bottleneck of the work emerges from the inability of the work to show which herbicides have been focused to show impact of changed climatic variables.

Conclusion: It is fine and gives clear insights on the topic.

Author Response

Abstract: the background is presented well and objectives are clear, however, the authors did not mention about the methods or approach, even  though it is a review, but must narrate the coverage of review in terms of scope, time period and nature of literature reviewed along with source. In addition, 1-2 sentences on key findings could also make it attractive/insightful for the journal readers. Keywords also need a revisit as per my understanding.

Ans: As this is a review paper, the previous literature from 1990 to 2021 were covered that show the effect of climate change on weed species and germination pattern. Revised as per suggestion.

Introduction: 

It is well-presented and contextualized. The background of the topic along with justification and nature of scope of the work is effectively taken up in this section. Section 2.1 needs a revisit as it has been overlooked to write its name. Both the sections 2 and 3 are a sort of results and discussion but as pointed out earlier, it would have been better if authors could clarify the selection of studies highlighted/referred in these sections. One would be more interested in the selection of literature more systematically rather than randomly. However, the selected literature is relevant and suited to the topic of the paper. One of the main bottleneck of the work emerges from the inability of the work to show which herbicides have been focused to show impact of changed climatic variables.

Ans:  So as previously stated we tried to cover published research article from past 40-40 years , infant to see how climates changing can also effect weed plants. So, for the herbicide efficacy we mainly talked about the activity of foliage applied herbicides might be altered by decreased stomatal conductance. Furthermore, we talked about most used glyphosate herbicides activity with changing environmental conditions. It would be lengthy to cover all herbicides activity, so we discussed general part and then one two examples.

Conclusion: It is fine and gives clear insights on the topic.

Reviewer 4 Report (New Reviewer)

General comments

This paper presents a well written detailed review of the potential effects of climate change on weed competition and herbicidal efficacy. Verify species names, e.g. in Table 2 replace virdis by viridis and crusgalli by crus-galli. Do not overuse species abbreviations e.g. the control of C arvensis and A. viridis…, when the binomial name have been stated in the previous paragraphs, the reader has to go back in the text to figure out what species is mentioned. Unless the binomial name is stated in the 1-2 previous sentences, restrain from not indicating the genus. I do not like the statement “evolving themselves”, evolution is not an intrinsic factor nor are weeds capable of intrinsic selection. The conclusion paragraph needs to be re-written.

Minor comments

Line 45 aggravates troublesome weed control…delete “troublesome”, weed control is rarely presented as easy or effortless so adding troublesome does not add value to the statement

Line 46 the crop yield, delete “the”

Line 54 the weed growth, delete “the”

Lines 59-70 plants leaves, replace by plant leaves…but what about plant roots in the case of soil applied herbicides? Rephrase the sentence

Line 70 replace XXXX by photosynthetic cycles?

Line 97…dominate competition, replace by be more competitive

Table 2…Amaranthus viridis, Echinochloa crus-galli

Line 110-114 This paragraph on migration needs to be re-written to be more weed specific…Although most weed propagules do not have structures that facilitate dispersal, human-mediated dispersal and dispersal by birds after ingestion are very important .. having weed seed mixed is often stated as contamination …of seed, feed, manure

Table 3, replace virdis by viridis, crusgalli by crus-galli

Line 134…was more as compared to, replace by was higher compared to

Line 142…evolving themselves….replace by something like …climate induced selection will favor weed populations more than crops

Line 234, has been studied previously…delete previously

Line 243, grown of soybean, replace by soybean growth

Line 300, the multiple resistant plants are not resistant to cyhalofop-butyl I suppose, clarify

Line 379-381 The improvements or reductions with the adjuvants is significant (I suppose?) but it it not high,  rephrase.

Line 385.. reported decreased in efficacy, replace by reported a decrease in efficacy

Lines 414-429 This conclusion is confusing …mentions the evolving themselves statement and does not really summarise the text.

Author Response

Reviewers 4:

Verify species names, e.g. in Table 2 replace virdis by viridis and crusgalli by crus-galli.

(Corrected as per suggestion)

Do not overuse species abbreviations e.g. the control of C arvensis and A. viridis…, when the binomial name have been stated in the previous paragraphs, the reader has to go back in the text to figure out what species is mentioned. Unless the binomial name is stated in the 1-2 previous sentences, restrain from not indicating the genus. I do not like the statement “evolving themselves”, evolution is not an intrinsic factor nor are weeds capable of intrinsic selection. The conclusion paragraph needs to be re-written.

Ans: revised as per suggestion and corrected species name.

Minor comments

Line 45 aggravates troublesome weed control…delete “troublesome”, weed control is rarely presented as easy or effortless so adding troublesome does not add value to the statement

Deleted troublesome as per suggestion

Line 46 the crop yield, delete “the”

Deleted as per suggestion

Line 54 the weed growth, delete “the”

deleted

Lines 59-70 plants leaves, replace by plant leaves…but what about plant roots in the case of soil applied herbicides? Rephrase the sentence

(Rephrased)

Line 70 replace XXXX by photosynthetic cycles?

Added as per suggestion

Line 97…dominate competition, replace by be more competitive

Replaced as per suggestion

Table 2…Amaranthus viridis, Echinochloa crus-galli

Corrected

Line 110-114 This paragraph on migration needs to be re-written to be more weed specific…Although most weed propagules do not have structures that facilitate dispersal, human-mediated dispersal and dispersal by birds after ingestion are very important .. having weed seed mixed is often stated as contamination …of seed, feed, manure

Revised and deleted extra part

Table 3, replace virdis by viridis, crusgalli by crus-galli

corrected

Line 134…was more as compared to, replace by was higher compared to

corrected

Line 142…evolving themselves….replace by something like …climate induced selection will favor weed populations more than crops

Added as per suggestion

Line 234, has been studied previously…delete previously

deleted

Line 243, grown of soybean, replace by soybean growth

replaced

Line 300, the multiple resistant plants are not resistant to cyhalofop-butyl I suppose, clarify

clarified

Line 379-381 The improvements or reductions with the adjuvants is significant (I suppose?) but it it not high,  rephrase.

rephrased

Line 385.. reported decreased in efficacy, replace by reported a decrease in efficacy

corrected

Lines 414-429 This conclusion is confusing …mentions the evolving themselves statement and does not really summarise the text.

Deleted evolving and rephrased

Round 2

Reviewer 3 Report (New Reviewer)

Paper can be accepted in its present form.

This manuscript is a resubmission of an earlier submission. The following is a list of the peer review reports and author responses from that submission.

Round 1

Reviewer 1 Report

This article deals with a very important and current issue related to the phenomenon of global warming in the context of the management of weed control (regulation of weed infestation). It is particularly important in the situation of faster adaptation of weeds (plasticity of these species) to changing climatic conditions than of cultivated plants. It should be presumed that in connection with the above, in the longer or near future, weeds will become even more competitive to cultivated plants (especially those in which the possibility of using herbicides is limited or impossible - e.g. some species of herbal plants). Moreover, the few studies conducted so far show that herbicides show greater herbicidal effectiveness under changing climatic conditions than when the climate was more predictable. The vast majority of scientific reports on this subject present the thesis that herbicides are less effective in the conditions of global warming.

The authors of this manuscript presented a thorough and comprehensive analysis of the issues they selected. The prospect of creating models of climate change (temperature, precipitation and soil type) in studying the variability of weed biology, and even their genome, seems to be very interesting. The same applies to research on the modification of the action of various groups and active substances of herbicides, or the development of new herbicides. This is a very important issue, because while striving to increase the effectiveness of herbicides, it is necessary to maintain their low phytotoxicity (care for the quality of the natural environment).

To sum up, the article is very well written, both in terms of content and style (language). Optionally, in Conclusions, a mention may be made of the problem of the increasing competitiveness of weeds (as a result of climate warming) in crops where we do not use herbicides (or we use them in a very limited way). What predictions can we make in this case? Perhaps research should be carried out on the greater genetic resistance of such crops to unfavorable factors of the habitat, so that they can more effectively compete with the weed flora? These are my sample suggestions for the authors to consider.

Reviewer 2 Report

The manuscript entitled "Impact of futuristic climate variables on weed biology and herbicidal efficacy: a review", is a review paper about the effects of climate change on weed biology and chemical control. The review deals with an interesting and actual topic, that will be more and more important in the near future. About weed management, unfortunately, the authors focused only about the effects on herbicides, thus limiting the appeal of this paper. Discussing the influence of climate change on various weed management methods would have been of more interest for both the scientific community and readers.

Moreover, on one side many sections report well-known aspects of weed biology and weed science without adding the last findings in bio-physiology. For this reason, I think that the novelty level of the first part is low.

On the other side, I found low comments provided by authors and low interconnections between the number of addressed topics. Overall, a common thread is missing in my opinion.

Please find below some specific comments and suggestions:

·       I think the paper would be more potent if reduced by about 20%. There is bit of redundancy in the text (please see some examples in the following comments)

·       Overall, the review appears too much descriptive over several parts, with very few critical speculations. The added value for the scientific community is therefore low, in my opinion

·       lines 29-33: there some repetitions in presenting the goals of the review. Please be more systematic. In addition, abstracts lacks of the main findings, which I suggest presenting via a bulleted list

·       line 34: I suggest changing the last keyword, it is too long and not attractive. “Herbicides” could be another keyword

·       references are not reported following the Journal’s guidelines. Nevertheless, they are often outdated

·       Nothing is reported about the methodology adopted to carry out this review. Bibliography is composed by 104 references in total, which seems a quite low number for a review article, suggesting strict criteria for literature selection. On which bases the article were chosen? How did you perform the bibliographic search? Please address these and other aspects concerning the methodology and report them in the text (generally at the end of the introduction)

·       I suggest deleting the sub-levels 2.1.1 and 2.2.2. These sub-paragraphs could be merged and written as only 2.1. Moreover, they could be highly summarised to avoid redundant and well-known information, also considering the presence of Table 1

·       Table 2: major weeds according to which criteria? Please specify in table caption. In addition, please add the taxonomist in all binomial names

·       lines 110-129: this paragraph deals with very well-known aspects of weed science. Please summarises or delete

·       Quality of presentation and resolution of figures could be improved

·       Conclusions are not consistent since they often refer to weed management practices, while the authors addressed only chemical control.

Reviewer 3 Report

Within this work the influence of climate changes were investigated that will likely have serious implications on agriculture production through its effects on food and feed crop yield and quality, forage and livestock production, and pest dynamics including troublesome weed control.

It is very important to investigate if the significance of the claim that the climate changes are actually influential to agronomy/weed parameters. For instance, are there statistically significant differences between data presented in Fig. 2 (Biomass, Seed)?

Thematically the work is interesting for the researchers and professionals and the proposed manuscript is relevant to the scope of the journal.

I found it appropriate for the Agronomy journal, but only after some modifications and clarification from the Authors.
The title is a clear representation of the manuscript's content. The abstract reflects realistically the substance of the work.

The overall organization and structure of the manuscript are appropriate. The paper is well written and the topic is appropriate for the journal.
The aim of the paper is well described and the discussion was well approached, its results and discussion are correlated to the cited literature data.
In the introductory part, the authors give elaboration of the overall context stating the motivation and the objectives of the work, literature review of the research pathways .
The literature review is comprehensive and properly done. However, most of the literature references are older than 10 years?

The novelty of the work must be more clearly demonstrated. Most of the literature references are older than 10 years? The significance of the Work: Given the large number of analyzed data, this is an interesting study with a possible significant impact in this area.

Statistical interpretation of the analytical data must be more properly presented. It is very important to investigate if the significance of the claim that the climate changes are actually influential to agronomy/weed parameters. 

Other Specific Comments: The work is properly presented in terms of the language. The work presented here is very interesting and well done, it is presented in a compact manner.